# Energy Conversion and Transfer in the Luminescence of CeSc_3_(BO_3_)_4_:Cr^3+^ Phosphor

**DOI:** 10.3390/ma16031231

**Published:** 2023-01-31

**Authors:** Lei Chen, Yabing Wu, Qi Liu, Yanguang Guo, Fanghai Liu, Bo Wang, Shizhong Wei

**Affiliations:** 1School of Materials Science and Engineering, Hefei University of Technology, Hefei 230009, China; 2National Joint Engineering Research Center for Abrasion Control and Molding of Metal Materials & Henan Key Laboratory of High-Temperature Structural and Functional Materials, Henan University of Science and Technology, Luoyang 471003, China; 3Intelligent Manufacturing Institute of HFUT, Hefei 230051, China; 4School of Applied Physics and Materials, Wuyi University, Jiangmen 529020, China

**Keywords:** light-emitting diodes (LEDs), near-infrared (NIR) phosphor, CeSc_3_(BO_3_)_4_:Cr^3+^, huntite, photoluminescence, energy transfer

## Abstract

Novel near-infrared (NIR) phosphors are in demand for light-emitting diode (LED) devices to extend their suitability for new applications and, in turn, support the sustainable and healthy development of the LED industry. The Cr^3+^ has been used as an activator in the development of new NIR phosphors. However, one main obstacle for the Cr^3+^-activated phosphors is the low luminescence efficiency due to the spin-forbidden d-d transition of Cr^3+^. The rare-earth (RE) huntite minerals that crystallize in the form of REM_3_(BO_3_)_4_ (M = Al, Sc, Cr, Fe, Ga) have a large family of members, including the rare-earth scandium borates of RESc_3_(BO_3_)_4_. Interestingly, in our research, we found that the luminescence efficiency of Cr^3+^ in the CeSc_3_(BO_3_)_4_ host, whose quantum yield was measured at 56%, is several times higher than that in GdSc_3_(BO_3_)_4_, TbSc_3_(BO_3_)_4_, and LuSc_3_(BO_3_)_4_ hosts. Hereby, the energy conversion and transfer in the luminescence of CeSc_3_(BO_3_)_4_:Cr^3+^ phosphor were examined. The Stokes shift of electron energy conversion within the Cr^3+ 4^T_2g_ level for the emission at 818 nm and excitation at 625 nm in CeSc_3_(BO_3_)_4_ host was evaluated to be 3775.1 cm^−1^, and the super-large splitting energy of the ^2^F_5/2_ and ^2^F_72_ sub-states of the Ce^3+^ 4f^1^ state, about 3000 cm^−1^, was found in CeSc_3_(BO_3_)_4_ host. The typical electronic thermal vibration peaks were observed in the excitation spectra of CeSc_3_(BO_3_)_4_:Cr^3+^. On this basis, the smallest phonon energy, around 347.7 cm^−1^, of the CeSc_3_(BO_3_)_4_ host was estimated. Finally, the energy transfer that is responsible for the far higher photoluminescence of Cr^3+^ in CeSc_3_(BO_3_)_4_ than in other hosts was proven through the way of Ce^3+^ emission and Cr^3+^ reabsorption.

## 1. Introduction

Light-emitting diodes (LEDs) have achieved big success in lighting and display applications [1,2]. To support the sustainable and healthy development of the huge LED industry, which ranges from substrate epitaxy, LED chip fabrication, and LED package to end-use applications, it is urgent to open up new fields of application [1,2]. For this aspect, the red and near-infrared LEDs show a bright future for applications in infrared photography, imaging, photomedicine, biomedicine, sensors, detectors, photosynthetic agriculture, and as a backlight for testing instruments [3,4,5,6,7].

Compared with multiple-chip LEDs, the phosphor-converted LEDs have many distinct advantages in terms of an easily tunable emission wavelength, low cost of manufacturing, and a short period research and development cycle. Moreover, the technique of fabricating blue LED chips is relatively more mature than that of other kinds of ultraviolet, near ultraviolet, and deep ultraviolet chips. In addition, the instruments used to grow blue LED chips, mainly metal–organic chemical vapor deposition (MOCVD), are abundant, and the key raw materials for fabricating blue LED chips are available. Nevertheless, due to the large Stokes shift, it is a challenge to convert the blue emission of LED chips into red and near-infrared photons by using phosphors. Both Eu^2+^ and Ce^3+^ have been widely used as activators to obtain yellow, green, and red emissions, but it is very hard to obtain near-infrared emissions from Eu^2+^ and Ce^3+^ under the excitation of blue light [8,9,10]. Comparatively, the Cr^3+^, whose d orbital has a relatively small crystal field splitting energy, is promising for obtaining red and near-infrared emissions under blue excitation [5,6,7]. However, the d-d transition of Cr^3+^ is spin-forbidden, and accordingly, the absorbance of Cr^3+^ is low, which results in the luminescence of the Cr^3+^-activated phosphor with poor efficiency. Recent progress achieved in the NIR phosphors activated by Cr^3+^ is described in the latest review papers on references [11,12,13]. Generally, the absorbance and luminescence efficiency of the Cr^3+^-activated phosphors are far below those of the Eu^2+^- and Ce^3+^- activated phosphors. To enhance the luminous efficiency of the Cr^3+^-activated phosphors, the absorbance of the phosphor should be improved above all.

The rare-earth (RE) scandium borates RESc_3_(BO_3_)_4_ have a calcite-derived huntite structure, crystallized in the trigonal R32 space group [14,15,16]. In the crystal lattice of RESc_3_(BO_3_)_4_, the Sc atom provides an ideal site for the activator Cr^3+^ because of their similar ion radii (Sc^3+^, 0.81; Cr^3+^, 0.69) and charge balance. Surprisingly, we found out that the luminescence efficiency of CeSc_3_(BO_3_)_4_:Cr^3+^ is several orders of magnitude higher than that of GdSc_3_(BO_3_)_4_:Cr^3+^, TbSc_3_(BO_3_)_4_:Cr^3+^, and LuSc_3_(BO_3_)_4_:Cr^3+^. This exceptional phenomenon mechanism was revealed in this work with energy conversion and transfer in the luminescence of CeSc_3_(BO_3_)_4_:Cr^3+^ phosphor.

## 2. Experimental Procedure

The phosphors RE(Sc_0.975_Cr_0.025_)_3_(BO_3_)_4_ (RE = Ce, Gd, Tb, and Lu), abbreviated as RESc_3_(BO_3_)_4_:Cr^3+^ below, were synthesized with a two-step solid-state reaction. The high-purity CeO_2_ (99.99%, Kepu, Ganzhou), Gd_2_O_3_ (99.99%, Kepu, Ganzhou), Tb_4_O_7_ (99.99%, Kepu, Ganzhou), Lu_2_O_3_ (99.99%, Kepu, Ganzhou), Sc_2_O_3_ (99.99%, Kepu, Ganzhou), H_3_BO_3_ (99.8%, Sinopharm), and Cr(NO_3_)_3_·9H_2_O (99.95%, Luoen, Shanghai) were used as raw materials. Firstly, the stoichiometric raw materials were thoroughly ground with a high-speed vibration ball mill. Next, the ball-milled mixtures were pre-fired at 500 ℃ for 2 h, and then the pre-fired products were ground again using the agate mortar and pestle. Finally, the phosphors were synthesized at 1250 ^o^C for 8 h in ambient air. Crystal structures of the phosphors were examined with the X-ray diffractometer (PANalytical, X-Pert PRO MPD) using a Cu target. The morphology was tested by scanning electron microscopy (JEOL, JSM-6490LV). The excitation and emission spectra of the samples were tested by a fluorescence spectrometer (Hitachi, F4600). The test sample was loaded into a self-made sample tank, pressed, and scraped flat to guarantee that the excitation and emission spectra were measured under the same conditions. The absorption spectra were collected with a UV–visible near infrared spectrophotometer (Shimadzu Corporation, Kyoto City, Japan, Shimadzu, UV-3600). All samples’ excitation and emission spectra were tested after the samples were synthesized and preserved for 8 months. The fluorescence quantum yield was measured with an absolute photoluminescence quantum yield measurement system (Hamamatsu Photonics K. K., Hamamatsu City, Janpan, Hamamatsu, Quantaurus-QY plus C13534-31), and the accumulation wavelength range is 300–1600 nm. The band gap, reflective spectra, absorption spectra, and the optical parameter of refractive index were calculated using the CASTEP module of Materials Studio 2019 software by employing the crystal structure of Ce_3_Sc(BO_3_)_4_ with PDF # 38-0720 as the initial structure model. In calculating the properties of RE’Sc_3_(BO_3_)_4_ (RE’ = Gd, Tb, Lu), the Ce atom was replaced with Gd, Tb, and Lu, respectively. Before performing computations, the crystal structure was optimized with the generalized gradient approximation (GGA).

## 3. Results and Discussion

The compounds RESc_3_(BO_3_)_4_ (RE = Ce, Gd, Tb, Lu) are members of a large group of huntite-family rare-earth borates REM_3_(BO_3_)_4_ (M = Al, Sc, Cr, Fe, Ga) that crystallize in the structure of huntite minerals, CaMg_3_(CO_3_)_4_ [14,15,16]. From the X-ray diffraction (XRD) patterns of the RESc_3_(BO_3_)_4_:Cr^3+^ (RE = Ce, Gd, Tb, Lu) phosphors shown in Figure 1a, we can see that the main diffraction peaks of CeSc_3_(BO_3_)_4_:Cr^3+^ phosphor are consistent with standard PDF# 38-0720 of CeSc_3_(BO_3_)_4_. When Ce is replaced with Gd, Tb, and Lu, the phosphors of GdSc_3_(BO_3_)_4_:Cr^3+^, TbSc_3_(BO_3_)_4_:Cr^3+^, and LuSc_3_(BO_3_)_4_:Cr^3+^ exhibit a similar diffraction pattern to that of CeSc_3_(BO_3_)_4_:Cr^3+^, as Ce, Gd, Tb, and Lu belong to the same rare-earth family elements. Furthermore, the minor peaks indicate the presence of minor intermediate compounds of TbBO_3_ and ScBO_3_, as well as the remaining unreacted CeO_2_ raw material. Indeed, the melting point of B_2_O_3_, about 450 ^o^C, is very low, while the melting point of rare-earth oxides is far higher. During the synthesis process of RESc_3_(BO_3_)_4_, the rare-earth orthogonal borates REBO_3_ usually appear first, followed by the formation of RESc_3_(BO_3_)_4_. For this reason, we always find that borates such as TbBO_3_ and ScBO_3_ always accompany the RESc_3_(BO_3_)_4_ phases. The three-dimensional structure of CeSc_3_(BO_3_)_4_, shown in Figure 1b, exhibits non-centrosymmetry with the space group R32 in the trigonal system. In the crystal lattice of CeSc_3_(BO_3_)_4_, both Sc and Ce are sixfold coordinated, as shown in Figure 1b. Considering ionic radium and charge balance, Cr^3+^ should take the place of Sc in the crystal lattice of RESc_3_(BO_3_)_4_. Except for pure hosts, the concentration of Cr^3+^ doped in each RESc_3_(BO_3_)_4_ sample was nominally fixed at 2.5 atm% of Sc in the crystal lattice. Thus, the luminescence properties of Cr^3+^ in the RESc_3_(BO_3_)_4_ host could be explained with the Tanabe–Sugano diagram as shown in Figure 1c.

The scanning electron microscope (SEM) image presented in Figure 2 shows that the particles of CeSc_3_(BO_3_)_4_:Cr^3+^ phosphor have a typical size of 2–10 μm, consisting of several small grains that have agglomerated together. Nonetheless, the characteristic trigonal profile of the grains could still be identified from the irregular particles, as indicated by the arrow in Figure 2.

The emission and excitation spectra of RESc_3_(BO_3_)_4_:Cr^3+^ (RE = Ce, Gd, Tb, Lu) displayed in Figure 3 show that the luminescence intensity of CeSc_3_(BO_3_)_4_:Cr^3+^ is nearly four times higher than that of LuSc_3_(BO_3_)_4_:Cr^3+^ and more than 10 times higher than that of GdSc_3_(BO_3_)_4_:Cr^3+^ and TbSc_3_(BO_3_)_4_:Cr^3+^. This phenomenon is very interesting. The measured quantum yield and absorbance of CeSc_3_(BO_3_)_4_:Cr^3+^ are 65.8% and 24.4%, respectively.

Figure 3a presents the emission spectra of RESc_3_(BO_3_)_4_:Cr^3+^ (RE = Ce, Gd, Tb, Lu) under the excitation of 471 nm, in which the broadband emission is attributed to the ^4^A_2_-^4^T_2_(^4^F) transition of Cr^3+^. By monitoring the emission of Cr^3+^ at 822, 830, and 866 nm, the excitation spectra of RESc_3_(BO_3_)_4_:Cr^3+^ (RE = Ce, Gd, Tb, Lu) are given in Figure 3b–d, respectively, in which two strong excitation bands in regions of 400–575 nm and 575–800 nm are attributed to the electron transitions of ^4^A_2_-^4^T_1_(^4^F) and ^4^A_2_-^4^T_2_(^4^F), respectively [11,12,13]. Moreover, minor excitation bands in the region of 200–400 nm are observed, which should be caused by host excitation (as discussed below) in addition to the high-level excitation of Cr^3+^ with the ^4^A_2_-^4^T_1_(^4^G) transition. Many electronic vibration peaks were observed in Figure 3b–d, particularly in Figure 3d as indicated by arrows, which for one side suggests the presence of strong thermal vibration of the crystal lattice on Cr^3+^ luminescence and for the other side suggests that the crystal lattice of RESc_3_(BO_3_)_4_:Cr^3+^ (RE = Ce, Gd, Tb, Lu) has an un-rigid structure. Through the electronic vibration peaks displayed in Figure 3d, the phonon energy of CeSc_3_(BO_3_)_4_:Cr^3+^ phosphor could be evaluated, in which the reciprocal of peak wavelengths corresponds to energy. According to quantum mechanics, the difference between two neighboring peaks corresponds to the energy of coupled phonons, nћω, i.e., n times the smallest one ћω, where n is the natural number. Hereby, the smallest phonon energy ћω, which is about 347.7 cm^−1^, was evaluated from the neighboring peaks at 718.3 nm and 736.7 nm as marked with pink arrows in Figure 3d. Using the relation of phonon energy of E = *ħ*ω, where *ħ* = h /2 π (h is the Planck constant, h = 6.626 × 10^−34^ J·s = 4.1369 × 10^−15^ eV·s; *ħ* is the reduced Planck constant, *ħ* = 1.055 × 10^−34^ J·s = 6.582 × 10^−16^ eV·s) and ω = 2 πν (radians per unit time), the angular frequency ω was evaluated at 4.26 × 10^21^ (rad/ S).

To reveal the mechanisms of the far higher photoluminescence of CeSc_3_(BO_3_)_4_:Cr^3+^ phosphor than others, the emission and excitation spectra of RESc_3_(BO_3_)_4_:Cr^3+^ (RE = Ce, Gd, Tb, Lu) were first normalized. After normalizing the emission intensity, the spectra in Figure 4a show that the emission peaks red shift upon the change of rare-earth elements in the order of Ce-Lu-Gd-Tb. However, the variation in emission peak is not evident. With the strongest excitation at 471 nm normalized to 1.0, the excitation spectra upon the emission at 822 nm are shown in Figure 3b. As indicated by the line in Figure 4b, the relative excitation intensity in the 550–700 nm region in the order of Tb-Gd-Lu-Ce is opposite to that presented in Figure 3a for luminescence red shift. To some extent, this phenomenon suggests that the relaxation of electrons from the ^4^T_1_(^4^F) to ^4^T_2_(^4^F) level that produces NIR emission results in the low-efficiency luminescence of the phosphors.

The element Lu, due to the inert electronic structure of [Xe]4f^14^, is widely used as a component for phosphor hosts, such as the green phosphor of Lu_3_Al_5_O_12_:Ce^3+^ for white LEDs [17]. The element Gd^3+^, which has a 4f semi-complete orbital electronic structure, i.e., [Xe]4f^7^, is not only used as a component of phosphor hosts but is also frequently used as a sensitizer, such as a phosphor (Y,Gd)BO_3_:Eu^3+^ for plasma display panels. Gd^3+^ is a good innate sensitizer due to the large energy difference between the ground state, ^8^S, and the lowest excited level, ^6^P. Thus, Gd^3+^ could play the role of an intermediate to transfer energy through the sublattice [18]. Ce^3+^ and Tb^3+^ are commonly used as phosphor activators to obtain various color emissions, but they can also be used as components of hosts, such as the traditional green phosphor LaPO_4_:Ce^3+^,Tb^3+^ for fluorescent lamps and the yellow phosphor Tb_3_Al_5_O_12_:Ce^3+^ for white LEDs [19,20]. Herein, the rare-earth elements Ce, Gd, Tb, and Lu were all used as the components of hosts for the activator Cr^3+^. Surprisingly, the photoluminescence of Cr^3+^ in the CeSc_3_(BO_3_)_4_ host is far higher than that in GdSc_3_(BO_3_)_4_, TbSc_3_(BO_3_)_4_, and LuSc_3_(BO_3_)_4_ hosts. To further reveal the mechanisms, the emission, excitation, and absorption spectra of the CeSc_3_(BO_3_)_4_ host without doping Cr^3+^ were studied.

By exciting the CeSc_3_(BO_3_)_4_ host with two different wavelengths of 323 and 358 nm, nearly the same configuration emission spectra were obtained, as shown in Figure 5a, which confirms that the emissions originate from the same mechanism. By monitoring the emission at 450 and 472 nm, the excitation spectra are shown in Figure 5b, in which two excitation peaks were observed. In Figure 5b, the two excitation peaks should originate from the 4f–5d transition of Ce^3+^. The spin–orbital coupling effect on the outer one electron in the 4f orbital splits the ground state of the Ce^3+^ electron into two states, ^4^F_5/2_ and ^4^F_7/2_. Thus, the two excitation peaks at 323 and 358 nm in Figure 5b were attributed to the ^4^F_5/2_–5d and ^4^F7_/2_–5d transitions of Ce^3+^. Accordingly, the emission spectra of the CeSc_3_(BO_3_)_4_ host presented in Figure 5a were attributed to the 5d–4f transition of Ce^3+^. By plotting the emission and excitation spectra of the CeSc_3_(BO_3_)_4_ host together, Figure 5c shows that there is almost no overlap between them, suggesting the CeSc_3_(BO_3_)_4_ host has a weak crystal field. The excitation peaks of the ^4^F_5/2_–5d and ^4^F_7/2_–5d transitions at 323 and 358 nm, respectively, in Figure 5b, show that the energy difference between the ^4^F_5/2_ and ^4^F7_/2_ states is about 3000 cm^−1^, which is equivalent to 9–10 phonons (as above-obtained, 347.7 cm^−1^) in CeSc_3_(BO_3_)_4_:Cr^3+^ phosphor. The energy difference between the ^4^F_5/2_ and ^4^F7_/2_ states in CeSc_3_(BO_3_)_4_:Cr^3+^ phosphor is far higher than that in the rigid-structure phosphors such as Y_3_Al_5_O_12_:Ce^3+^ [17], also indicating the soft structure of CeSc_3_(BO_3_)_4_.

The absorption spectrum of the CeSc_3_(BO_3_)_4_ host is shown in Figure 5d, in which a broadband consisting of several peaks was observed. By plotting the absorption and the excitation spectra of CeSc_3_(BO_3_)_4_ together, the absorption peaks of the characteristic ^4^F_5/2_–5d and ^4^F7_/2_–5d transitions of Ce^3+^ were easily identified. The extra absorption within 200–300 nm may be caused by the electron transitions within the Sc-O or Sc-BO_3_ bonds. Therefore, Figure 5d suggests that the minor excitation band within 200–400 nm should be caused by the CeSc_3_(BO_3_)_4_ host, but the ^4^A_2_-^4^T_1_(^4^P) transition of Cr^3+^ cannot be excluded.

Figure 5d shows that the low energy cutoff of the absorption spectrum is approximately 400 nm, suggesting the band gap of the CeSc_3_(BO_3_)_4_ host is about 3.1 eV. The calculated band gap of CeSc_3_(BO_3_)_4_ shown in Figure 6a is about 0.494 eV. The result is comparable with the calculated value of 0.317 eV of CeSc_3_(BO_3_)_4_ band gap shared in the Materials Project with ID mp-16097 [21]. Due to the inherent nature of the first principle calculation, the calculated band is usually smaller than the experimental one [22]. Nevertheless, the calculated band still could provide some important information for understanding the mechanism. Figure 6d shows that the Fermi level consists of a line, which should originate from the 4f orbital of CeSc_3_(BO_3_)_4_ and is consistent with the experimental results in Figure 5d. The refractive index is an intrinsic parameter that determines the optical behavior of reflection and absorption. As the result shown in Figure 6b, the refractive index changes with photon energy. However, the blue excitation and NIR emission have photon energies less than 5 eV. As shown in Figure 6c,d, there is a significant difference in the reflection and absorption spectra among the REc_3_(BO_3_)_4_ hosts within the energy range of 5–50 eV. Yet, there is almost no difference in the energy below 5 eV, which further indicates that Ce plays a special role in improving the luminescence efficiency of CeSc_3_(BO_3_)_4_:Cr^3+^.

By plotting the emission of the CeSc_3_(BO_3_)_4_ host and the excitation of the CeSc_3_(BO_3_)_4_:Cr^3+^ phosphor together, the spectra are shown in Figure 7. In Figure 7, a good match between the emission of the CeSc_3_(BO_3_)_4_ host and the ^4^A_2_-^4^T_1_(^4^F) excitation of Cr^3+^ in the CeSc_3_(BO_3_)_4_:Cr^3+^ phosphor suggests that there is an energy transfer from the CeSc_3_(BO_3_)_4_ host Ce^3+^ to the activator Cr^3+^. With the intensity normalized, the overlap of Cr^3+^ absorption and Ce^3+^ emission is 99% of Ce^3+^ emission, suggesting an efficient pathway from Ce^3+^ emission to Cr^3+^ re-absorption.

However, the excitation spectra in Figure 3b–d show that the energy transfer from the host to Cr^3+^ is not efficient since the relative intensity of excitation in the 200–400 nm range is very low. Moreover, no overlap between the emission and excitation spectra of Ce^3+^ suggests that the Stokes shift is very high. As a result, the energy transfer between variant Ce^3+^ ions is almost impossible. This phenomenon could be explained by the low phonon energy in CeSc_3_(BO_3_)_4_. Thus, the energy transfer pathway from Ce^3+^ to Cr^3+^ through electron resonance effect was excluded. By combining with the absorption spectrum in Figure 5d, we get to know that the energy transfer from Ce^3+^ to Cr^3+^ in the CeSc_3_(BO_3_)_4_:Cr^3+^ phosphor under the excitation of blue light is thus through coaction of Ce^3+^ emission and Cr^3+^ reabsorption. Accordingly, the mechanism of the energy transfer from the CeSc_3_(BO_3_)_4_ host to the Cr^3+^ activator and energy conversion processes in the CeSc_3_(BO_3_)_4_:Cr^3+^ phosphor could be depicted with Figure 1.

## 4. Conclusions

In summary, the luminescence efficiency of the CeSc_3_(BO_3_)_4_:Cr^3+^ phosphor is 3–20 orders of magnitude higher than that of GdSc_3_(BO_3_)_4_:Cr^3+^, TbSc_3_(BO_3_)_4_:Cr^3+^, and CeSc_3_(BO_3_)_4_:Cr^3+^ phosphors. The measured quantum yield and absorbance of CeSc_3_(BO_3_)_4_:Cr^3+^ are 65.8% and 24.4%, respectively. The characteristic peaks of electronic vibrations were observed in the excitation spectra of CeSc_3_(BO_3_)_4_:Cr^3+^, on which the smallest energy of the CeSc_3_(BO_3_)_4_ phonons was evaluated at 347.7 cm^−1^. The splitting energy of the ^2^F_3/2_ and ^2^F_3/2_ sub-states of the Cr^3+^ 4f^1^ state in CeSc_3_(BO_3_)_4_ was determined at 3000 cm^−1^, and the Stokes shift of electrons within the emission (818 nm) and excitation (625 nm) states of the Cr^3+^ 4T_2g_ level was 3775.1 cm^−1^. All these parameters suggest that CeSc_3_(BO_3_)_4_ has a super-soft structure. The self-emission of the CeSc_3_(BO_3_)_4_ host, which originates from the Ce^3+^ 5d–4f transition, was confirmed. However, the Stokes shift between the ground state and the excited states of Ce^3+^ in the CeSc_3_(BO_3_)_4_ host is too big, resulting in the excitation energy hardly being able to transfer through the electron resonance effect among variant Ce^3+^ ions. Therefore, the energy transfer that is responsible for the intensified luminescence of Cr^3+^ in the CeSc_3_(BO_3_)_4_ host was mainly through the Ce^3+^ emission and Cr^3+^ reabsorption. This work sheds light on developing new highly efficient Cr^3+^-activated phosphors.

## Data Availability

For any data request, please contact the corresponding authors.

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
