# Peer review of "Energy Conversion and Transfer in the Luminescence of CeSc3(BO3)4:Cr3+ Phosphor"

_materials, 2023, doi:10.3390/ma16031231_

Round 1

Reviewer 1 Report

The present work describes photoluminescence properties of Cr-doped CeSc3(BO3)4. The material is new, and presented results are interesting. I can recommend a publication after some minor modifications listed below.

1.       In the introduction, “but it is very hard to get near-infrared emissions from Eu2+ and Ce3+ under the excitation of blue light [8-10]. Comparatively, the Cr3+ is a suitable activator to get red and near-infrared emissions, due to its small splitting energy of d orbital in the crystal field [5-7].”, the logic is unclear. For Eu and Ce, it is difficult to obtain red/NIR emission under blue excitation. But you do not point out whether the blue excitation is possible in Cr3+ or not.

2.       In 2, please comment Cr concentration.

3.       In Fig 1 (a), some unidentified peaks are observed in Tb- and Ce-based ones. What are they?

4.       In the caption of Fig.1 (c), Cr and Cr3+ co-exist.

5.       In line 103, 2-10 m?

6.       For the PL quantum yield, please explain the accumulation wavelength range. Although you only comment about Ce-one, and how about others?

7.       In Fig. 3 (d), for the calculation of phonon energy, which peak do you use for the calculation? Since I’m not unfamiliar with such as calculation (I generally use Raman for the phonon energy), and would you like to explain?

8.       In the caption of Fig. 5, “(c) excitation, (c) emission and excitation together,”, (c) overlap.

9.       For the Ce-one, how do I consider about the bandgap energy? ~360 nm (Fig. 5) or ~400 nm (Fig. 6)?

10.    Would you like to comment the position of this work? I mean, for example, if you compare with other NIR phosphors by Cr, how is the emission efficiency?

Reviewer 2 Report

This manuscript presents the luminescence properties of CeSc3(BO3)4:Cr3+ phosphor. Authors reported that the luminescence efficiency of the CeSc3(BO3)4:Cr3+ is 3-20 times that of GdSc3(BO3)4:Cr3+, TbSc3(BO3)4:Cr3+, and LuSc3(BO3)4:Cr3+. In the manuscript, the author also attempts to explain the mechanism of energy transfer in CeSc3(BO3)4:Cr3+. The research topic itself is very interesting and has great general impact, making it of interest to readers in the field. However, questions remain about the accuracy of the research methodology. The results are also complex and very difficult to understand, and it is difficult to determine from the data in the manuscript alone whether they provide a correct explanation of the energy transfer. I recommend that editor reconsider whether to publish after a major revision.

About the methodology:

-Luminescence evaluation

In intensity comparisons by PL, there are many factors that affect the quantitative nature of PL measurements, such as sample placement, sample size, and temperature on the day of measurement. If the measurement device you used can guarantee the quantifiability of PL intensity, please mention that in your paper. However, my personal opinion is that you should evaluate and compare the luminescence efficiency of the samples using absolute PL quantum yield, which can be evaluated more quantitatively. In fact, the authors state in the methods chapter that they measured absolute PL quantum yield, but they do not provide any results or discussion about them. They should show them.

-preparation of the phosphors

Please list the source of purchase of the raw materials used and the purity of the raw materials.

Phosphors are sintered at 1250 degC. Please explain how you determined this temperature.

About the result and discussion:

-energy transfer

In the discussion, the authors state that the energy transition efficiency from Ce3+ to Cr3+ is very low. On the other hand, authors mention that the energy transfer which is responsible for the intensified luminescence of Cr3+ in CeSc3(BO3)4 host was mainly through the emission of Ce3+ and re-absorption of Cr3+. That is probably the wrong statement. Since there is no absorption in the absorption spectrum of the CeSc3(BO3)4 host, Ce3+ should not be contributing to any emission for the blue excitation light around 471 nm. (Rather, Cr3+ is easily affected by the surrounding crystal field due to the change of host, which may be a factor, but I think the current data is insufficient for the discussion.)

-XRD

The XRD peak of GdSc3(BO3)4:Cr3+ appears to be split. What is the cause of the split peak?

-phonon energy

Please describe in detail how “phonon energy” is calculated in your manuscript.

-absolute PL quantum yield

Again, I recommend discussing luminescence efficiency by comparing absolute PL quantum yield, not PL intensity.

Other comments:

Misspelling are scattered throughout the manuscript. Please correct them.

Reviewer 3 Report

The authors make an in-deep analysis of the luminescence efficiency of Cr3+ in CeSc3(BO3)4 host. The energy transfer responsible for the intensified photoluminescence of Cr3+ in CeSc3(BO3)4 host was assigned to Ce3+ emission and Cr3+ re-absorption.

The phosphors were synthesized, and their crystal structure was examined by X-ray diffractometry.  The morphology was examined by SEM. Emission and absorption spectra proved stability of the phosphor after months.

The revision of the manuscript is required before acceptance and my comments are below

-Novelty of the work must be discussed and underlined in the Introduction. The authors contribute is not so clear. Inherent recent literature must be added.

-As the fluorescence quantum yield was used to quantify the phenomenon and to compare the effects, more space should be given to the description of the method.

-Discussion (sect.4) should be improved by adding a minimum of theoretical analysis about the energy transfer from host to Cr3+ . A synergy between experimental data and periodic and molecular DFT calculations adds completeness and value to the work. In this regard, I recommend taking a cue and citing an inherent work, https://doi.org/10.1002/ejic.201400095.

-English in the manuscript must be improved, a general detailed revision is to be recommended due to typos and grammar mistakes

Round 2

Reviewer 2 Report

The authors revised the manuscript very well in response to almost all my comments. 

However, I still do not understand the explanation of "That is to say, by exciting the CeSc3(BO3)4 host with blue light like 471 nm, the electrons undergo rapid relaxation from the excited state to the ground state, and meanwhile, the emission spectra were broadened due to the thermal vibration of the crystal lattice. Under the excitation of blue, therefore, the energy transfer from Ce3+ to Cr3+ in the CeSc3(BO3)4:Cr3+ phosphor is through co-action of the emission of Ce3+ and re-absorption of Cr3+, which can explain the enhanced luminescence of CeSc3(BO3)4:Cr3+, nearly several times that of GdSc3(BO3)4:Cr3+, TbSc3(BO3)4:Cr3+, and LuSc3(BO3)4:Cr3+, in Fig. 2a. " in the manuscript. 

We do not know how the Ce3+ ions in the host of CeSc3(BO3)4 contribute to the emission enhancement of the blue emission at 471 nm, the same wavelength as the Ce3+ emission (not excitation). Can you really make the above discussion from just the data obtained? If this is a general phenomenon in the field of phosphor research, please refer to previous studies describing similar phenomena. Alternatively, measuring PL decay for blue excitation and luminescence at Cr3+ for all hosts might provide some useful information.

Reviewer 3 Report

The manuscript has been sufficiently improved to warrant publication in Materials. No other comments 

Author Response

Thanks to the the reviewer so much for your great effiort paid on our manuscript.